# Inoculum-Based Dosing: A Novel Concept for Combining Time with Concentration-Dependent Antibiotics to Optimize Clinical and Microbiological Outcomes in Severe Gram Negative Sepsis

**DOI:** 10.3390/antibiotics12111581

**Published:** 2023-10-31

**Authors:** Alwin Tilanus, George Drusano

**Affiliations:** 1Department of Infectious Diseases, Clinica Los Nogales, Calle 95 # 23-61, Bogota 110221, Colombia; 2Institute for Therapeutic Innovation, University of Florida, 6550 Sanger Road, Orlando, FL 32827, USA; george.drusano@medicine.ufl.edu

**Keywords:** PK, PD, dose optimization, time-dependent antibiotics, concentration-dependent antibiotics, inoculum effect, resistance suppression

## Abstract

Certain classes of antibiotics show “concentration dependent” antimicrobial activity; higher concentrations result in increased bacterial killing rates, in contrast to “time dependent antibiotics”, which show antimicrobial activity that depends on the time that antibiotic concentrations remain above the MIC. Aminoglycosides and fluoroquinolones are still widely used concentration-dependent antibiotics. These antibiotics are not hydrolyzed by beta-lactamases and are less sensitive to the inoculum effect, which can be defined as an increased MIC for the antibiotic in the presence of a relatively higher bacterial load (inoculum). In addition, they possess a relatively long Post-Antibiotic Effect (PAE), which can be defined as the absence of bacterial growth when antibiotic concentrations fall below the MIC. These characteristics make them interesting complementary antibiotics in the management of Multi-Drug Resistant (MDR) bacteria and/or (neutropenic) patients with severe sepsis. Global surveillance studies have shown that up to 90% of MDR Gram-negative bacteria still remain susceptible to aminoglycosides, depending on the susceptibility breakpoint (e.g., CLSI or EUCAST) being applied. This percentage is notably lower for fluoroquinolones but depends on the region, type of organism, and mechanism of resistance involved. Daily (high-dose) dosing of aminoglycosides for less than one week has been associated with significantly less nephro/oto toxicity and improved target attainment. Furthermore, higher-than-conventional dosing of fluoroquinolones has been linked to improved clinical outcomes. Beta-lactam antibiotics are the recommended backbone of therapy for severe sepsis. Since these antibiotics are time-dependent, the addition of a second concentration-dependent antibiotic could serve to quickly lower the bacterial inoculum, create PAE, and reduce Penicillin-Binding Protein (PBP) expression. Inadequate antibiotic levels at the site of infection, especially in the presence of high inoculum infections, have been shown to be important risk factors for inadequate resistance suppression and therapeutic failure. Therefore, in the early phase of severe sepsis, effort should be made to optimize the dose and quickly lower the inoculum. In this article, the authors propose a novel concept of “Inoculum Based Dosing” in which the decision for antibiotic dosing regimens and/or combination therapy is not only based on the PK parameters of the patient, but also on the presumed inoculum size. Once the inoculum has been lowered, indirectly reflected by clinical improvement, treatment simplification should be considered to further treat the infection.

## 1. Introduction

To maximize the chance of surviving severe sepsis, bundles have been implemented, which include timely administration of antibiotics (ideally within one hour) and aggressive fluid resuscitation.

The Surviving Sepsis Guidelines (SSG) 2021 recommend: “For patients with sepsis induced hypoperfusion or septic shock at least 30 mL/ kg of IV crystalloid fluid should be given within the first 3 h of resuscitation” [1]. Sub-therapeutic levels of antibiotics are a major concern in critically ill patients [2,3] and are particularly related to increased Volume of distribution (Vd) and changes in renal function, including augmented renal clearance (ARC) [4]. Especially the Vd of hydrophilic antibiotics (including beta-lactams and aminoglycosides) is likely to be increased in the critically ill patient [5]. In addition, Vd seems to be proportional with disease severity [6,7] and aggressive fluid management in the early phase of the disease process will likely expand the Vd even more, resulting in serum “dilution of antibiotics” [8] and prolong the half-life (t_1/2_) of the antibiotic. Adequate dosing of antibiotics in a patient with a high inoculum infection, elevated Vd and additional (renal) losses is challenging and requires an individualized approach. The Surviving Sepsis Guidelines 2021 also state: “For adults with sepsis or septic shock and high risk for multidrug resistant (MDR) organisms, we suggest using two antimicrobials with Gram-negative coverage for empiric treatment over one Gram-negative agent.” without further specifications [1]. The cornerstone of therapy in severe sepsis is a beta-lactam administrated by “prolonged infusion of beta-lactams for maintenance (after an initial bolus)” [1]. 

Beta-lactam antibiotics act by acylating penicillin-binding proteins (PBP) in a time-dependent manner throughout the dosing interval, which can only be minimally accelerated by increasing the dose [9]. Therefore, a combination with a concentration-dependent antibiotic (e.g., aminoglycoside, fluoroquinolone) could serve to quickly lower the inoculum, create PAE (temporal bacterial growth cessation), and possibly reduce PBP expression, maximizing clinical/microbiological outcomes. High inoculum infections and subtherapeutic antibiotic exposure at the site of infection have been associated with inadequate resistance suppression and adverse clinical/microbiological outcomes [10]. 

Although the inoculum size can be measured in vivo [11], it can also be indirectly estimated by disease severity. Higher inocula will likely present with a more severe illness and will likely require less time to yield positive blood cultures. 

In the early phase of severe sepsis, rapid reduction of the inoculum size mediated by dose optimization is a priority. 

Frequently, clinicians tend to make dosing considerations solely based on PK characteristics of the patient (e.g., renal function, volume of distribution, etc.), but not merely on PD characteristics (e.g., type of suspected organism(s), inoculum size, mechanism of resistance, etc.). In this article, the authors therefore propose a novel concept of “Inoculum Based Dosing” in which the antibiotic dosing regimens (e.g., loading dose, high dose administered by prolonged infusion) and/or combination therapy with a concentration-dependent antibiotic are based on the presumed inoculum size. With clinical improvement, one can indirectly assume that the inoculum has been lowered and dosing adjustments can be considered. 

## 2. Methods: Search Strategy and Selection Criteria

Combinations of the search terms “PK” (Pharmacokinetics), “PD” (Pharmacodynamics), “Dose”, “concentration dependent”, “aminoglycosides”, “amikacin”, “gentamicin”, “renal toxicity”, “ototoxicity”, “fluoroquinolones”, “ciprofloxacin”, “Inoculum effect”, “post-antibiotic effect”, ODD (Once Daily Dosing), “AUC” (area under the curve), “MIC” (minimal inhibitory concentration), and “resistance suppression” were used for multiple searches in Pubmed. Relevant (English) articles published from 1970 until 2023, including pre-clinical studies, RCTs, observational studies, meta-analyses, and systematic reviews, were evaluated and selected for this article. A total of 86 publications that fulfilled the inclusion criteria were included in this article.

## 3. Concentration-Dependent Antibiotics

Concentration-dependent antibiotics show clear concentration-dependent killing associated with interference in protein or nucleic acid synthesis. The PK-PD indices for concentration-dependent antibiotics are generally described as the maximal concentration (Cmax) divided by MIC (Cmax/MIC) and the area under the curve (24 h) divided by MIC (AUC_24h_/MIC) [12,13]. 

## 4. Aminoglycosides

Aminoglycosides are among the oldest antibiotics used in clinical practice to date, which started with the introduction of streptomycin in 1944 and amikacin in 1972. The most recent aminoglycoside, plazomycin, was approved by the FDA in 2018, but unfortunately is not available anymore [14].

Aminoglycosides are polycationic molecules that electrostatically bind to negatively charged compounds on the outer membrane of Gram-negative bacteria. After transport across the bacterial cytoplasmic membrane through voltage-gated channels, they subsequently bind to the 30S subunit of ribosomes and induce a conformational change in the A-site of the 16S rRNA, resulting in mistranslation of proteins [15,16]. 

## 5. Aminoglycosides and the Post-Antimicrobial Effect (PAE)

PAE is an in vitro phenomenon that refers to a period of time (after the complete removal of an antimicrobial) during which there is no growth of the bacterium being exposed. PAE has been observed in most antimicrobial agents, both in Gram-positive and Gram-negative bacteria.

PAE appears to be influenced by multiple factors such as type of organism, type/concentration of the antibiotic, duration of antimicrobial exposure, antimicrobial combinations, inoculum, culture conditions, and leucocyte count [12,17,18]. Since aminoglycosides bind irreversibly to ribosomes, PAE may represent the time needed to synthesize new ribosomes [17]. Aminoglycosides possess a relatively long post-antibiotic effect (PAE) against Gram-negative bacteria, which is higher in vivo than in vitro and is exposure-dependent [17,19,20,21]. In addition, mouse model studies have shown that PAE appears to be several hours longer in non-neutropenic mice due to renal insufficiency and the strain being studied [22].

## 6. Aminoglycosides and the Inoculum Size

Few studies have been performed to examine the inoculum effect of aminoglycosides with inconsistent results, which are probably explained by variations in culture conditions and the type of organism being investigated. Yu and Kelly both reported in vitro inoculum effects of aminoglycosides against various Gram-negative bacteria [23,24]. However, Szabo et al. did not report an inoculum effect for amikacin in vivo (single dose 7.5 mg/kg) when studying a high inoculum of an ESBL-producing *Klebsiella pneumoniae* strain [25].

## 7. Clinical Studies with Aminoglycosides

Based on pre-clinical in vitro and in vivo (animal) infection models, the 24-h AUC/MIC ratio showed the best correlation between antimicrobial exposure and therapeutic efficacy [12]. However, in clinical trials, the Cmax/MIC appears to be the most precise PK-PD index to predict clinical outcome. Moore et al. [26] analyzed data from 236 patients with Gram-negative bacterial infections who were treated with aminoglycosides. A remarkably strong dose-response effect was found between an increasing peak concentration to MIC ratio (Cmax/MIC) and clinical response. Their study revealed that a PK-PD target Cmax/MIC > 8 was associated with a clinical response rate of more than 80%, and subsequent studies reconfirmed these findings [27,28]. In addition, resistance suppression appears to be best predicted by the Cmax/MIC index [28,29].

## 8. Once Daily versus Multiple Daily Dosings of Aminoglycosides

Traditionally, the use of aminoglycosides has been limited due to renal/ototoxicity issues, and multiple studies have been conducted to optimize their use. Limiting the dose of aminoglycosides to avoid toxicity carries a risk of poor target attainment (e.g., Cmax/MIC < 8; AUC/MIC ratio < 175) and an unfavorable outcome. Indeed, various studies have shown inadequate Cmax serum levels of aminoglycosides in ICU patients related to increased Vd [30,31,32]. Since aminoglycosides show a concentration-dependent PAE (associated with their irreversible binding to the 30S ribosomal subunit) and clinical outcome is strongly correlated with Cmax/MIC, once-daily dosing (ODD) became the principal mode of dosing for aminoglycosides. The optimal dose of aminoglycosides could be defined as having a high likelihood of a good therapeutic outcome while minimizing the likelihood of concentration-related drug toxicity [13,14,15,16,17,19,20,21,22,23,24,25,26,27,28,29,30,31,32,33]. Observational studies comparing ODD with multiple daily dosing (MDD) showed a clear tendency towards improved clinical outcome, a higher probability of target attainment, and a favorable nephro/otoxicity profile [34,35,36], which also applies to studies performed with neutropenic patients [37,38].

## 9. Aminoglycoside-Induced Renal Toxicity

It is estimated that up to 10–25% of patients treated with aminoglycosides develop nephrotoxicity, which depends on multiple variables. One observational study among geriatric patients showed that the duration of therapy (e.g., more than 7 days) is the single most important risk factor for nephrotoxicity [39]. Other risk factors include age (e.g., over 50 years), baseline creatinine, and dosing frequency (ODD versus MDD) [39,40]. Uptake of aminoglycosides in renal epithelial cells mainly occurs in an endocytosis-dependent manner and is mediated by a multi-ligand endocytic receptor named megalin [41]. Aminoglycosides can exert a nephrotoxic effect due to their accumulation in the proximal renal tubular cells, ultimately causing tubular obstruction, although vascular and glomerular disturbances are also involved in the pathogenesis [42]. There are very few publications that report on the reversibility of kidney function after aminoglycoside-induced acute kidney injury. Selby et al. reported that 78% of patients with gentamycin-induced nephrotoxicity had complete kidney function recovery at the time of hospital discharge [43]. Paquette et al. reported that only 51% of patients with gentamycin/tobramycin-induced nephrotoxicicty, had complete recovery of renal function. However, it should be noted that they included only patients who had received more than five days of therapy, which likely explains the poor complete renal function recovery rate [44]. 

## 10. Aminoglycoside-Induced Ototoxicity

Ototoxicity related to aminoglycosides is the most severe and irreversible type of toxicity, which can be divided into vestibular and cochlear toxicity. Aminoglycosides can generate free radicals within the inner ear, which can cause permanent damage to sensory cells and neurons, resulting in permanent hearing loss. Cochlear damage can produce permanent hearing loss, while damage to the vestibular apparatus results in vertigo, ataxia, and/or nystagmus [45]. Multiple studies have shown that cumulative exposure (AUC) and duration of therapy (not merely peak concentrations) are the main risk factors for ototoxicity [46,47,48], supporting the ODD regimen for a maximum of a few days.

## 11. Combination Therapy: Aminoglycosides and Beta-Lactams

Pre-clinical in vitro studies have shown promising results with beta-lactam/aminoglycoside combinations resulting in synergistic bacterial killing. It was postulated that changes in the cell wall caused by the effects of beta-lactam antibiotics facilitated the uptake of aminoglycosides [49,50]. However, the opposite (e.g., aminoglycosides facilitating the uptake of beta-lactams) was not observed but could possibly occur if higher doses were used. Conflicting results were obtained in in vivo studies with poor correlations between in vitro synergistic testing and clinical outcome [51]. Paul et al. published a Cochrane review in 2014 analyzing studies of beta-lactam monotherapy versus beta-lactam plus aminoglycoside combination therapy for sepsis [52]. They included 69 trials, which corresponded to 7863 (non-neutropenic) patients hospitalized with urinary tract, intra-abdominal, skin, and soft tissue infections, pneumonia, and infections of unknown source. All-cause mortality rates were insignificantly different between the two groups, and combination treatment was associated with a significant risk of nephrotoxicity. The authors stated that the addition of an aminoglycoside to beta-lactams for sepsis should be strongly discouraged. 

Several observations can be made about the results and the studies included in this analysis. Firstly, only 22 trials (31.9%) compared the same beta-lactam in both study arms, which might have had an impact on clinical outcome. Secondly, the average time of treatment with aminoglycosides was 8.6 days, which at least in part explains the high rate of renal insufficiency observed. Thirdly, only one study (1.6%) used a once-daily high dose (amikacin 20 mg/kg/day). This study was performed with 74 patients diagnosed with ventilator-associated pneumonia, with 19 patients in the cefepime-amikacin group. Since the antimicrobial effect of amikacin is directly proportional to its peak concentration, the underrepresentation of high-dose amikacin studies in the meta-analysis likely has profound effects on the study outcome. Fourthly, in the only high-dose amikacin study analyzed, there was no information about the presence of shock. Information about disease severity is crucial since this is correlated with inoculum size, Vd, and/or the presence of ARC, which are the most important risk factors for underdosing and outcome. Fifthly, many beta-lactams used in the study were cephalosporins or penicillins, which, at least based on in vitro studies, are known to be at risk for an inoculum effect that could have affected clinical outcome. Sixthly, the included patients showed a high level of heterogeneity with respect to the type of infectious disease (e.g., pneumonia or abdominal, etc.), types of microorganisms involved, and their susceptibility patterns, which likely influenced the outcome of the meta-analysis as well. 

Based on the existing PK-PD studies with suboptimal dosing regimens and globally increasing rates of MDR organisms, without better alternatives, complete abandonment of aminoglycosides becomes a questionable issue. Le et al. reported a significant synergistic activity in vitro when meropenem was combined with amikacin against KPC-3-producing *K. pneumonia* isolates as compared to both medicaments alone [53], highlighting the potential role for amikacin in treating infections caused by KPC-producing organisms.

## 12. PK-PD Studies with Relatively High-Dose Aminoglycosides

Taccone et al. published in 2010 the results of a 25 mg/kg amikacin first loading dose in 74 patients with septic shock. Target attainment (Cmax/MIC > 8) was achieved in 70% of the patients, but no correlation between target attainment and clinical outcome was reported [54]. Subsequently, Galvez et al. reported in 2011 that a 30 mg/kg daily dose of amikacin in patients with severe sepsis or septic shock resulted in significantly higher target attainment (Cmax/MIC > 8) in 76% of patients as compared with the other groups (25 mg/kg and 15 mg/kg). Interestingly, the rate of nephrotoxicity was not different between the groups. In addition, the authors stated that there is no need to adjust the loading dose to renal function in patients with septic shock. Other clinical outcome parameters were not reported [55]. Subsequent PK-PD studies reported similar findings, with rates of target attainment (Cmax/MIC > 8) up to 80% but without a clear correlation between improved mortality and/or clinical outcome [56,57,58,59,60]. Possible explanations for the lack of correlation between outcome and mortality are numerous: heterogeneity of the study population (e.g., more severe disease is related to increased Vd and associated sub-therapeutic levels of amikacin), presence of ARC resulting in increased renal excretion, dose (e.g., 25 mg/kg versus 30 mg/kg), type of PK-PD index applied (CMax/MIC versus AUC/MIC), PK-PD target applied (e.g., Cmax/MIC > 8 versus Cmax/MIC > 10 or AUC/MIC > 175), and the impact of the MIC of the infecting organisms (e.g., slight increases in the MIC will have a major effect on the index outcome).

## 13. Fluoroquinolones 

Fluoroquinolones are broad-spectrum bactericidal antibiotics that act by inhibiting the activity of two key enzymes involved in DNA synthesis: DNA gyrase and topoisomerase IV [61,62]. Intracellular fluoroquinolone concentrations are the net result of diffusion-mediated drug uptake by porins and pump-mediated efflux [63]. Bacterial killing by fluoroquinolones is partly understood but appears to be biphasic, concentration- and quinolone-type-dependent. Relatively lower concentrations are associated with more rapid killing, but once the concentration exceeds a break point (“optimal bactericidal concentration”), the killing rates become slower. At a molecular level, lower concentrations of quinolones block DNA transcription by inhibiting gyrase and topoisomerase IV, and at higher concentrations, RNA and protein synthesis are inhibited [64,65].

The PK-PD index that best describes the relation between exposure and efficacy is AUC24h/MIC [12]. Similar to aminoglycosides, they show a PAE [18,66,67] and are relatively unaffected by higher inocula [68,69]. 

In addition, evidence suggests that fluoroquinolones can indirectly modulate the immune response, which seems to be mediated by the suppression/induction of multiple pro-inflammatory cytokines, ultimately resulting in the growth and activation of T and B lymphocytes [70].

Despite these interesting characteristics, the use of quinolones nowadays is greatly limited due to widespread resistance and toxicity issues. Resistance mechanisms of quinolones are complex and multifactorial but mainly mediated by target-site gene mutations that alter the drug-binding affinity of the target enzymes. In addition, reduced membrane permeability has been linked to quinolone resistance [62]. A detailed discussion with respect to the complex mechanisms of action and resistance is beyond the scope of this paper. 

## 14. PK-PD Studies with Fluoroquinolones

Forrest et al. performed a PK-PD study with 74 critically ill patients, mainly diagnosed with pneumonia, treated with varying doses of ciprofloxacin IV (200 mg q12 up to 400 mg q8h). 

Their results demonstrated a remarkable 80% probability of clinical and microbiologic cure in those patients with an AUC of at least 125 (Forrest et al. AAC 1993) [71]. Ambrose et al. reported a breakpoint for microbiological response of AUC24/MIC > 33.7 in 58 patients with pneumonia caused by Streptococcus pneumonia [72]. Subsequently, Drusano et al. published the results of a PK-PD study with 58 patients with nosocomial pneumonia, mainly caused by Gram-negative bacteria, treated with levofloxacin (750 mg IV/day). The authors reported a statistically significant breakpoint for the microbiological response of AUC24/MIC > 87 [73]. 

These pioneer PK-PD studies consistently demonstrated a critical breakpoint of the AUC/MIC and the chance of microbiological/clinical success, which at least in part seems to be influenced by the total daily dose and type of pathogen involved. When using conventional doses of ciprofloxacin (e.g., 400 mg q12h IV), a PK-PD target of AUC24h/MIC >125 is frequently not achieved [74,75]. 

Zelenitsky published a retrospective analysis of 178 cases of bloodstream infections due to Gram-negative organisms treated with ciprofloxacin. They reported that the risk of treatment failure was 27.8 times higher in those patients who did not achieve an AUC24/MIC ≥ 250, demonstrating the importance of using higher than conventional doses [76]. 

## 15. Combination Therapy: Fluoroquinolones and Beta-Lactams 

In vitro and in vivo studies combining beta-lactams with fluoroquinolones resulted in conflicting results [51]. In general, beta-lactams combined with fluoroquinolones rarely cause synergistic effects in enterobacterales, but interestingly, synergism has been reported against *Pseudomonas aeruginosa* [77,78,79,80]. Bliozitis et al. published a meta-analysis of eight randomized controlled trials of neutropenic patients treated with either a beta-lactam plus aminoglycoside or a beta-lactam plus fluoroquinolone. The authors reported comparable or better outcomes with the ciprofloxacin/beta-lactam combination versus an aminoglycoside/beta-lactam combination [81]. However, the heterogeneity of the patients, microorganisms involved, and dosing regimens likely affected the study outcomes.

## 16. Rationale for Combining Time and Concentration-Dependent Antibiotics in High Inoculum Infections

As mentioned in the introduction, the Surviving Sepsis Guidelines recommend empirical combination therapy of two antibiotics with Gram-negative coverage in patients with severe sepsis/septic shock and risk factors for MDR organisms. Beta-lactams are generally considered the backbone of empirical therapy in sepsis. However, these antibiotics show a time-dependent killing that is related to the acylation reaction in the serine active site of the PBPs, which requires sufficient time above the MIC and occurs in a dose-independent manner [9]. The minimal time required above the MIC to achieve bacterial stasis and bacterial kill depends directly on the PBP affinity of the beta-lactam and the degree of PBP saturation. Carbapenems have shown the highest rates of cellular penetration and PBP affinity in Gram-negative bacteria: maximal kill is achieved when the free drug concentrations remain above the MIC for at least 40% of the dosing interval, as compared to 50% for penicillin’s and 60–70% for cephalosporin’s, respectively [82]. However, time above the MIC is not the only factor that determines the efficacy of beta-lactams; also, the inoculum should be considered. With the exception of carbapenems, most of the beta-lactams are affected by the inoculum effect [83], which is more likely to occur in severe sepsis, exerting pressure on their efficacy in these critically ill patients. Early rapid target attainment has been linked to resistance suppression and is a determining factor for clinical and microbiological outcomes, especially in high-inoculum infections [10]. Although carbapenems require the least time above the MIC, they still need a considerable proportion of the dosing interval to achieve sufficient PBP acylation. When considering a dosing interval of every 8 h, 40% of the dosing interval translates to at least 3 h to achieve maximal bacterial kill. This critical time frame could be further optimized by combining it with a concentration-dependent antibiotic. To rapidly achieve target attainment as early as possible, we previously proposed a meropenem loading dose (1–2 g) followed within 1–2 h by a high maintenance dose administered by prolonged infusion in severe Gram-negative sepsis [84]. Theoretically, one could speculate that reducing the PBP expression of a bacterium by inhibiting its synthesis would facilitate the work of a beta-lactam as the total number of PBPs to be acylated is reduced. This effect could be further enhanced by quickly lowering the inoculum and creating PAE, preventing bacterial regrowth, and maximizing the likelihood of resistance suppression.

Furthermore, one could argue that the expression of other proteins, such as porins and efflux pumps, will also be reduced. The net result of the reduced expression of these two proteins could limit cellular uptake but also limit the expulsion of the aminoglycosides, which could be a reason to limit its use for a maximum of a few days. Daikos et al. reported that the first exposure of Gram-negative bacteria to an aminoglycoside antibiotic in vitro induces a biphasic bactericidal response and adaptive drug resistance with reduced cellular uptake of the drug, indirectly supporting this hypothesis [85].

Therefore, it would be reasonable to combine a time-dependent with a concentration-dependent antibiotic in high-inoculum infections presenting as severe sepsis. These findings could be considered a novel hypothesis to explain synergism in combination therapy (Figure 1 and Figure 2).

## 17. Combination Therapy: Beta-Lactam and Aminoglycosides

One option could be a beta-lactam combined with an aminoglycoside. Aminoglycosides have shown concentration-dependent bacterial killing mediated by ribosomal binding, a PAE of several hours, and (based on a few in vitro observations), their entry is facilitated by membrane disruption induced by beta-lactams. In addition, one could argue that by blocking the ribosomes, the synthesis and expression of PBPs and other proteins will be reduced, theoretically facilitating the action of the beta-lactams. However, at present, there are no data to support this hypothesis. To maximize the likelihood of target attainment (e.g., Cmax/MIC > 8–10) and to minimize toxicity, once daily high dose amikacin (25–30 mg/kg/day, maximally for a few days) should be considered in any patient with septic shock, especially neutropenic patients and/or those with risk factors for MDR organisms, independent of the renal function at baseline. The initial administration of an aminoglycoside should not merely be considered to achieve synergism or antibiotic spectrum amplification, but to quickly lower the inoculum and achieve PAE, creating optimal conditions for the beta-lactam. To our knowledge, there are no clinical studies that have examined the optimal moment of administration when these two antibiotics are combined. Based on scarce in vitro observations, one could argue that beta-lactam should be administered before the aminoglycoside. However, the opposite (uptake of beta-lactams facilitated by membrane disruptions caused by aminoglycosides) cannot be discarded when high-dose aminoglycosides are being used. A loading dose of a beta-lactam (ideally 1–2 g meropenem in 30 min) should be given, followed within 1 to 2 h by the administration of a high dose (e.g., 6 gr meropenem/day for patients with normal renal function or ARC) by prolonged infusion [84,85,86]. Considering the reduced half-life and ARC in the early phase of severe sepsis and to maximize the likelihood of synergism, one could argue to administer the (high-dose) aminoglycoside within approximately one hour after the administration of the beta-lactam. However, this assumption has not been studied in animal models. Once hemodynamic stability is achieved and/or culture results are obtained, deescalation and dose adjustments should be considered. 

## 18. Combination Therapy of a Beta-Lactam Combined with a Fluoroquinolone

When aminoglycosides are not an option, the combination of a beta-lactam with a fluoroquinolone could be considered, but this depends largely on local susceptibility patterns and the clinical scenario. 

To our knowledge, there are no clinical data to support first administering the fluoroquinolone followed by the beta-lactam, vice versa, or simultaneously. However, based on the mechanism of action (growth suppression related to inhibition of DNA/mRNA synthesis and immunomodulatory effects), one could argue to administer the fluoroquinolone first, shortly (e.g., within one hour), followed by the beta-lactam. Similar to the aminoglycosides, the initial administration of a fluoroquinolone should not merely be considered to achieve synergism or antibiotic spectrum amplification, but to quickly lower the inoculum and achieve PAE, creating optimal conditions for the beta-lactam. To maximize the likelihood of target attainment (e.g., AUC/MIC > 250) and minimize the risk of toxicity, higher than conventional doses (e.g., 400 mg q8 h in the case of ciprofloxacin or levofloxacine 750 mg/day) should be considered in any patient with severe septic shock, especially neutropenic patients. This also applies to those with risk factors for MDR organisms, independent of baseline renal function, for a maximum of a few days. Once hemodynamic stability is achieved and/or culture results are obtained, de-escalation and dose adjustments should be considered.

## 19. Conclusions

Beta-lactam antibiotics act by acylation and saturation of the PBP and will always require a minimal time above the MIC, which is unlikely to be influenced by increasing the dose. PD endpoints like bacterial stasis and killing vary among beta-lactam antibiotics and depend on PBP affinity and saturation. If a dosing interval of every eight hours is being applied, beta-lactam concentrations still need to be at least 3 h above the MIC to achieve bacterial killing when carbapenems are being considered. This time period is considerably longer for penicillin’s and cephalosporin’s, respectively, making them less favorable antibiotics in septic shock. To initiate the acylation reaction of the PBPs as fast as possible, a (meropenem) loading dose within 1–2 h, followed by a high maintenance dose (administrated by prolonged infusion), should be considered in severe (Gram-negative) sepsis. Without the support of a second concentration-dependent antibiotic, a possible unfavorable clinical outcome might be expected during this critical time frame in severe sepsis. Concentration-dependent antibiotics such as aminoglycosides or fluoroquinolones, directly or indirectly, will interfere with protein synthesis at the ribosome level, likely resulting in reduced expression of PBPs, porins, and efflux pumps. Additionally, concentration-dependent antibiotics will quickly lower the inoculum and create a PAE of a few hours during which bacterial growth is unlikely to occur. As a consequence, the time-dependent acylation and saturation of the PBPs by the beta-lactams will be optimized. By quickly lowering the inoculum and creating PAE, the clinical outcome is likely to be optimal, and resistance suppression can be achieved. Combination therapy should only be considered in critically ill (neutropenic) patients with presumed high inoculum infections and/or infections by MDR organisms. 

To minimize the risk of toxicity and maximize the likelihood of target attainment, once daily high dose amikacin (e.g., 25–30 mg/kg/day) for maximally a few days can safely be used, even in the setting of renal insufficiency, and is unlikely to deteriorate renal function or cause ototoxicity. To maximize the AUC, higher than conventional doses of fluoroquinolones for the shortest time possible can also be safely used. Once the inoculum is lowered, as reflected indirectly by clinical improvement, the concentration-dependent antibiotic should be suspended, and beta-lactam monotherapy in lower doses should be sufficient to further treat the infection.

## Figures and Tables

**Figure 1 antibiotics-12-01581-f001:**
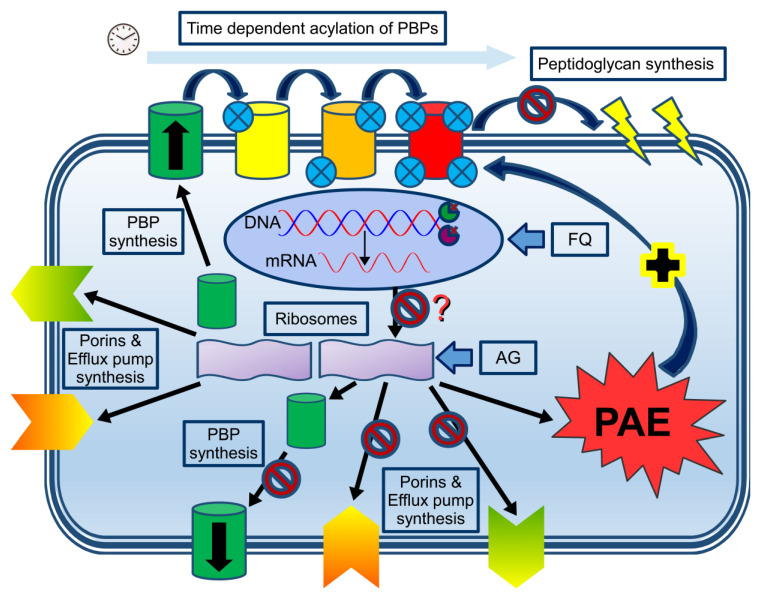
Combining time- and concentration-dependent antibiotics at the cellular and molecular level. DNA expression, resulting in mRNA (oval circle representing the nucleus), will synthesize PBP, porins, and efflux pumps in the non-blocked ribosome (left purple ribosome unit), which will be transported to the cell membrane. Subsequent time-dependent acylation of the serine active site (small blue circles representing the degree of saturation of the PBPs) will ultimately block peptidoglycan synthesis with subsequent membrane disruption (upper part of the graph). In the presence of aminoglycosides (AG) or fluoroquinolones (FQ), protein synthesis is likely to be blocked either directly by the AG (right purple ribosome unit) or indirectly (inhibition of DNA/mRNA synthesis as shown by the binding of the two key enzymes (Gyrase and Topoisomerase IV at the replication fork), although this hypothesis remains to be proven. Based on scarce in vitro data, inhibition of protein synthesis by fluoroquinolones seems to occur only when higher doses are being used. Inhibition of protein synthesis will likely result in reduced expression of PBPs, porins, and efflux pumps, although this concept has not been studied. The reduced expression of these proteins will likely reduce the cellular uptake/expulsion of aminoglycosides and fluoroquinolones. In addition, the bacterium will enter a PAE state with bacterial (re)growth inhibition, theoretically facilitating the workload of the beta-lactam antibiotics (dark blue arrow with the + sign).

**Figure 2 antibiotics-12-01581-f002:**
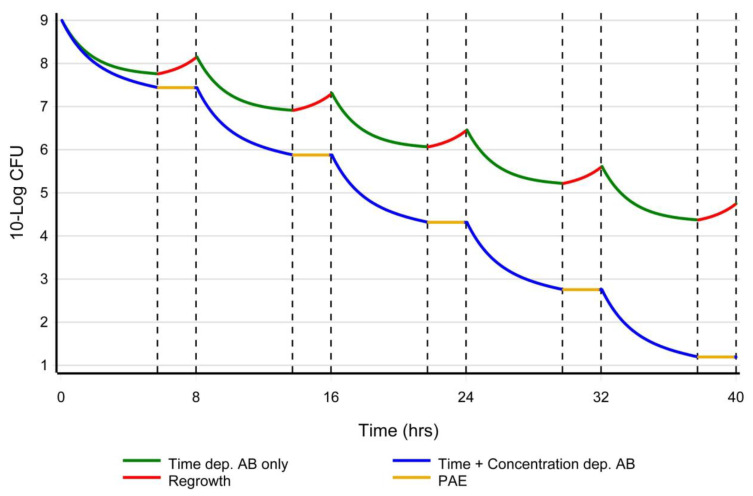
A hypothetical representation of inoculum-based dosing. Y-axis: log Colony Forming Units (CFU) as a function of time (x-axis). When only a (time-dependent) beta-lactam is used (green line), bacterial regrowth (upsloping of the curve between the dotted lines in red) is to be expected when the concentration of the antibiotic falls below the MIC. An exception are the carbapenems, which have shown a PAE of a few hours. When a concentration-dependent antibiotic (e.g., aminoglycoside or fluoroquinolone) is combined with a beta-lactam (blue line), a PAE of a few hours is to be expected when the concentration of the antibiotic falls below the MIC and bacterial (re)growth will not be observed (horizontal orange line of the curve between the dotted lines). A concentration-dependent antibiotic will quickly lower the inoculum, possibly reduce the bacterial PBP expression, and prevent bacterial regrowth during the PAE period, while in the meantime, beta-lactam can acylate and saturate the reduced number of PBPs in the shortest time possible, ultimately resulting in membrane disruption.

## Data Availability

Not applicable.

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
