# Peer review of "Inoculum-Based Dosing: A Novel Concept for Combining Time with Concentration-Dependent Antibiotics to Optimize Clinical and Microbiological Outcomes in Severe Gram Negative Sepsis"

_antibiotics, 2023, doi:10.3390/antibiotics12111581_

Round 1
Reviewer 1 Report
Comments and Suggestions for Authors
Comments:
1. The abstract provides a concise summary of the main topic of the article; however, the abstract could benefit from a brief statement regarding the significance or potential impact of the proposed "Inoculum Based Dosing" concept. This would help readers understand why this concept is novel and important in the context of antibiotic therapy.
2. Although the introduction effectively emphasizes the critical role of optimizing antibiotic treatment in sepsis, it would be advantageous to provide a more specific statement outlining the article's objectives and offering a brief preview of what readers can expect in the subsequent sections. This would offer a clearer roadmap for navigating the article.
3. The reference to the Surviving Sepsis Guidelines enhances the credibility of the discussion.
4. While the introduction successfully sets the stage for the concentration-dependent antibiotics discussion, explicitly stating the article's primary objectives or research questions would provide readers with a clearer understanding of the article's focus.
5. Inclusion of information about the use of English articles and the specified timeframe in the Search Strategy and Selection Criteria adds transparency to the search process.
6. The section contains various technical terms and abbreviations (e.g., PK, PD, MIC, AUC, ODD) that may require explanations for readers unfamiliar with these concepts. Providing concise definitions or explanations would enhance the accessibility of the text.
7. In some sections, references within sentences are used excessively. Consider consolidating these references and providing more concise explanations to improve readability.
8. The article effectively highlights the potential benefits of combination therapy with aminoglycosides and fluoroquinolones. However, it could present a more balanced view by discussing potential drawbacks and limitations, such as the risk of increased antibiotic resistance or adverse effects associated with combination therapy.
9. While the article briefly mentions resistance suppression as a benefit of combination therapy, expanding on this point by explaining how combination therapy contributes to resistance suppression and citing relevant studies or evidence would be beneficial.
10. While the article discusses theoretical concepts and mechanisms, incorporating more practical examples or case studies illustrating the application of combination therapy in clinical settings would provide real-world context for the discussed strategies.
11. Although the conclusion section effectively summarizes the main findings, it could be enhanced by explicitly restating the practical implications of the discussed concepts. Readers should be left with a clear understanding of how to apply this information in clinical practice.
Comments on the Quality of English Language
Moderate editing of English language required
Author Response
Dear Reviewer 1,
We would like to think you for the time taken to revise our manuscript and your valuable suggestions to improve the manuscript. We have tried our best to make the corresponding corrections in the manuscript.
Best regards,
Alwin Tilanus
George Drusano
Specific comments:
- The abstract provides a concise summary of the main topic of the article; however, the abstract could benefit from a brief statement regarding the significance or potential impact of the proposed "Inoculum Based Dosing" concept. This would help readers understand why this concept is novel and important in the context of antibiotic therapy.
Author response: A brief statement regarding the significance and potential impact of the proposed “inoculum-based dosing” concept has been added to the abstract.
- Although the introduction effectively emphasizes the critical role of optimizing antibiotic treatment in sepsis, it would be advantageous to provide a more specific statement outlining the article's objectives and offering a brief preview of what readers can expect in the subsequent sections. This would offer a clearer roadmap for navigating the article.
Author response: The introduction has been expended with more specific statements outlining the article’s objectives. One relevant reference has been added. - The reference to the Surviving Sepsis Guidelines enhances the credibility of the discussion.
Author response: The first reference of the article corresponds to the Surviving Sepsis Guidelines and, we agree with you, enhances the credibility of the discussion. - While the introduction successfully sets the stage for the concentration-dependent antibiotics discussion, explicitly stating the article's primary objectives or research questions would provide readers with a clearer understanding of the article's focus.
Author response: The introduction has been expended with more specific statements about the importance of the inoculum size in severe infections. In addition, a statement is made about how a high inoculum size together with inadequate antibiotic exposure in the early phase of severe sepsis could adversely affect clinical and microbiological outcome. - Inclusion of information about the use of English articles and the specified timeframe in the Search Strategy and Selection Criteria adds transparency to the search process.
Author response: The search strategy and selection criteria section has been expended. - The section contains various technical terms and abbreviations (e.g., PK, PD, MIC, AUC, ODD) that may require explanations for readers unfamiliar with these concepts. Providing concise definitions or explanations would enhance the accessibility of the text.
Author response: Unexplained abbreviations have now been described in the manuscript. - In some sections, references within sentences are used excessively. Consider consolidating these references and providing more concise explanations to improve readability.
Author response: After care evaluation of the manuscript, we believe that the references were not used excessively within sentences are were placed there were it is needed to support the text. - The article effectively highlights the potential benefits of combination therapy with aminoglycosides and fluoroquinolones. However, it could present a more balanced view by discussing potential drawbacks and limitations, such as the risk of increased antibiotic resistance or adverse effects associated with combination therapy.
Author response: Before describing the rationale for combination therapies (pages 9-11) a detailed discussion about dosing and toxicity issues related to aminoglycosides and fluoroquinolones are presented. We believe that we have clearly emphasized that combination therapy should be used only in high inoculum infections and/or patients with risk factors for MDR organisms, as short as possible to avoid toxicity and to suppress resistance. We also stated that in the case of aminoglycosides that reduced expression of porins could limit its uptake and therefore likely will reduce efficacy after a few days (Ref. 83: Daikos et al. AAC 1991). - While the article briefly mentions resistance suppression as a benefit of combination therapy, expanding on this point by explaining how combination therapy contributes to resistance suppression and citing relevant studies or evidence would be beneficial.
Author response: We deliberately focused in our article on a novel concept of how to quickly lower the inoculum and we provided the possibly involved mechanisms as summarized in figure 1 and 2.
Reduced expression of PBPs and creating PAE by the use of concentration dependent antibiotics could facilitate the work load of beta-antibiotics as the total number of PBPs to be acylated is reduced and bacterial regrowth is restricted during PAE.
We believe that further expending on mechanisms of resistance suppression would be beyond the scope of this paper and is still largely underexplored. - While the article discusses theoretical concepts and mechanisms, incorporating more practical examples or case studies illustrating the application of combination therapy in clinical settings would provide real-world context for the discussed strategies.
Author response: Based on preclinical in vitro / in vivo (animal) studies and clinical PK-PD studies discussed in the article, we tried to define patient groups that could benefit from short durations of (high dose) combination therapies. These groups would be “any patient with severe septic shock, especially neutropenic patients and/or those with risk factors for MDR organisms” as mentioned in page 10. However, we agree in that the clinical trials are needed to confirm the postulated concepts. - Although the conclusion section effectively summarizes the main findings, it could be enhanced by explicitly restating the practical implications of the discussed concepts. Readers should be left with a clear understanding of how to apply this information in clinical practice.
Author response: The conclusion section has been adjusted. It states that the decision for combination therapy (beta-lactam + concentration dependent antibiotic) should be considered only in critically ill (neutropenic) septic patients with high inoculum infections for the shortest duration possible (maximally a few days) to lower the inoculum, not to treat the entire infection. Once clinical improvement is observed (e.g. the inoculum is lowered) beta-lactam monotherapy should be installed.
Reviewer 2 Report
Comments and Suggestions for Authors
General Comments:
The article addresses the issues of „concentration dependent“ (higher doses result in increased bacterial killing) and „time dependent “antimicrobial activity of certain classes of antibiotics (time that antibiotic concentrations remain above the MIC). Since the great number of multi-drug-resistant Gram negative bacteria are still susceptible to aminoglycosides (and significantly less to fluoroquinolones) they are still antibiotics of choice for treatment of certain health problems. The article is important, interesting, the topic is challenging and up-to-date. Included are numerous interesting evaluations (and conclusions) of studies and their results. The rationale for combining time and concentration dependent antibiotics in high inoculum infections is well explained.
There are only several issues that need to be checked. After addressing these issues, I recommend it for publishing in your Journal.
Specific Comments:
Page 1, Lines 19-20. Please explain or rephrase „depending on the susceptibility breakpoint being applied“
Page 1, Line 27. Please be more specific with „PBP“
Page 2, Lines 63-75. It is a general rule to describe abbreviation when they are used for the first time in text. The authors should consider adding missing information.
Pages 5,6 Lines 189, 243, 245, 263. It is not mandatory, but it is useful to include certain „time-info“ when describing research of other authors. It is included in references, but the time context might also be useful.
Page 6, Lines 282-283. Confusing sentence. Consider revising.
Page 7, Lines 327-329. Confusing sentence. Consider revising.
Figure 2. It is nice presentation of presented topic. Authors should consider adding some (provisional) points on horizontal/vertical axis
Page 10, Lines 411-413. Confusing sentence. Consider revising.
Comments on the Quality of English LanguageMinor editing of English language required (see comments).
Author Response
Dear Reviewer 2,
We would like to thank you for the time taken to revise our manuscript and your valuable suggestions to improve the manuscript. We have tried our best to make the corresponding corrections in the manuscript.
Best regards,
Alwin Tilanus
George Drusano
Specific Comments:
Page 1, Lines 19-20. Please explain or rephrase „depending on the susceptibility breakpoint being applied“
Author response: The sentence has been modified and clarified (e.g. CLSI or EUCAST susceptibility breakpoints).
Page 1, Line 27. Please be more specific with „PBP“
Author response: The abbreviation PBP has now been described (Penicillin Binding Protein).
Page 2, Lines 63-75. It is a general rule to describe abbreviation when they are used for the first time in text. The authors should consider adding missing information.
Author response: All the abbreviations used in the manuscript are now described.
Pages 5,6 Lines 189, 243, 245, 263. It is not mandatory, but it is useful to include certain „time-info“ when describing research of other authors. It is included in references, but the time context might also be useful.
Author response: Maybe you can provide us with more information regarding the “time-info” you wish to include? After careful revision of the lines, we have decided not to make changes.
Page 6, Lines 282-283. Confusing sentence. Consider revising.
Author response: We revised the line but it is not clear to us why the sentence is confusing. We stated that the efficacy of beta-lactams not only depends on the time the concentrations remain above the MIC and that other factors such as inoculum should also be considered: Higher inocula could reduce the beta-lactam susceptibility.
Page 7, Lines 327-329. Confusing sentence. Consider revising.
Author response: This section discusses the combination therapy of beta-lactams with aminoglycosides and the optimal moment to administer each medicament to optimize their effect. Based on scarce in vitro data, membrane disruptions induced by beta-lactams could facilitate the aminoglycoside uptake. However, the opposite (uptake of beta-lactams facilitated by membrane disruptions caused by aminoglycosides) cannot be discarded if higher doses of aminoglycosides would have been used.
The sentence has been adjusted.
Figure 2. It is nice presentation of presented topic. Authors should consider adding some (provisional) points on horizontal/vertical axis
Author response: Figure 2 has been adjusted and now has CFU values and hours on the time axis.
Page 10, Lines 411-413. Confusing sentence. Consider revising.
Author response: We revised the line but we believe it is well written. We summarize that the use of once daily (high dose) aminoglycosides will enhance the likelihood of target attainment, will quickly lower the inoculum and is unlikely to cause renal/ototoxicity.
Reviewer 3 Report
Comments and Suggestions for Authors
Dear authors,
Thanks for this manuscript. Gram negative sepsis is an increasingly prevalent disease, with many controversies about optimal abx management. Therefore, this review is absolutely relevant. Yet, I suggest the following changes in order to enhance the quality:
Major
- The concept of inoculum effect is not defined in the introduction, please clarify that for the readers who are potentially not familiar with this. The same goes up for the concept of PAE.
- The aim of the current review remains vague- please enlighten that!
- Lines 133-152: please elaborate on the reversibility of the renal toxicity.
Minor
- Abstract lines 11-13: would write ‘higher concentrations result in’ opposed to ‘higher doses result in.’
- Fluoroquinolones are not a good example of concentration dependent abx given that these exhibit time dept features too. Metronidazole is a better example.
- Lines 15-17: Please write ‘aminoglycosides are not hydrolyzed by …” instead of concentration dependent abx.
- Lines 49-52: please clarify that these recommendation pertains empiric therapy, rather than double coverage in case of an identified MDR organism. Also reflect that in lines 271-273.
Comments on the Quality of English LanguageNo concerns.
Author Response
Dear Reviewer 3,
We would like to thank you for the time taken to revise our manuscript and your valuable suggestions to improve the manuscript. We have tried our best to make the corresponding corrections in the manuscript.
Best regards,
Alwin Tilanus
George Drusano
Specific Comments:
Major
- The concept of inoculum effect is not defined in the introduction, please clarify that for the readers who are potentially not familiar with this. The same goes up for the concept of PAE.
Author response:
- The inoculum effect and PAE are now well described in the introduction.
- The aim of the current review remains vague- please enlighten that!
Author response:
- The aim of this review has been further highlighted.
- Lines 133-152: please elaborate on the reversibility of the renal toxicity.
Author response:
- A small discussion about the reversibility of aminoglycoside induced renal toxicity is added.
- Two relevant references have been added.
Minor
- Abstract lines 11-13: would write ‘higher concentrations result in’ opposed to ‘higher doses result in.’
Author response: The word “Higher doses” has been replaced by “higher concentrations” as requested.
- Fluoroquinolones are not a good example of concentration dependent abx given that these exhibit time dept features too. Metronidazole is a better example.
Author response: The PK-PD index of fluoroquinolones is the AUC/MIC and therefore it is indeed true that these antibiotics exhibit time dependent features as well. However, when we look at the molecular mechanisms, the concentration dependent features predominates when it comes to efficacy (Lewin et al. Eur J Clin Microbiol infect. 1991). The efficacy of metronidazole is also described by AUC/MIC.
- Lines 15-17: Please write ‘aminoglycosides are not hydrolyzed by …” instead of concentration dependent abx.
Author response: The word “concentration dependent” has been replaced by “these antibiotics…”, referring to the previous sentence as requested.
- Lines 49-52: please clarify that these recommendation pertains empiric therapy, rather than double coverage in case of an identified MDR organism. Also reflect that in lines 271-273.
Author response: Both lines state the word empirical therapy. In the sections on page 10-11 the authors clarify that the combination therapy is not merely to amplify the antibiotic spectrum, but to rapidly reduce the inoculum possibly by reducing PBP expression and creating PAE.
Once the inoculum is lowered, de-escalation to beta-lactam monotherapy should be applied to further treat the infection.
Round 2
Reviewer 1 Report
Comments and Suggestions for Authors
Thank you for your diligent efforts in addressing the comments raised in the initial review. Your revisions have significantly improved the manuscript's overall quality and clarity. I appreciate the thoroughness with which you addressed the specific issues.
Author Response
Dear Reviewer 1,
Thank you for your revision.
We are pleased to read that our adjustments sufficiently addressed your observations.
Best regards,
Alwin Tilanus
Reviewer 3 Report
Comments and Suggestions for Authors
Thanks for this revision.
Although I really liked reading the revision, the concept of inoculum based dosing will be new to readers. This concept has not been outlined sufficiently in the introduction: what is inoculum based dosing, appreciate that this is a hypothetical concept. The current statement in the introduction is way to generic: "In this article the authors will shed a different light on how to optimize the use of concentration dependent antibiotics (limited to aminoglycosides and fluoroquinolones) alone or in combination with beta-lactams, especially in the early phase of severe high 74 inoculum infections."
Therefore, please define the IOD, the underlying assumptions, and how you explore this ahead in the text.
Author Response
Dear Reviewer 3,
Thank you for your revision and suggestions.
We fully revised the abstract and introduction with special attention to the new concept of "Inoculum Based Dosing". We believe it is now better defined and explained as requested.
Best regards,
Alwin Tilanus
Round 3
Reviewer 3 Report
Comments and Suggestions for Authors
Thanks for this clarification and amendments to the text!